# The Effects of E-Cigarette Vapor Components on the Morphology and Function of the Male and Female Reproductive Systems: A Systematic Review

**DOI:** 10.3390/ijerph17176152

**Published:** 2020-08-24

**Authors:** Kamila Szumilas, Paweł Szumilas, Anna Grzywacz, Aleksandra Wilk

**Affiliations:** 1Department of Physiology, Pomeranian Medical University, 70-111 Szczecin, Poland; kamila.szumilas@pum.edu.pl; 2Department of Social Medicine and Public Health, Pomeranian Medical University, 71-210 Szczecin, Poland; pawel.szumilas@pum.edu.pl; 3Independent Laboratory of Health Promotion, Pomeranian Medical University, 70-204 Szczecin, Poland; grzywacz.anna.m@gmail.com; 4Department of Histology and Embryology, Pomeranian Medical University, 70-111 Szczecin, Poland

**Keywords:** e-cigarettes, vapors, male and female reproductive systems

## Abstract

E-cigarettes, a comparatively new phenomenon, are regarded as a safer alternative to conventional cigarettes. They are increasingly popular among adolescents of both sexes, and many smokers use e-cigarettes in their attempts to quit smoking. There is little understanding of the effects of exposure to e-cigarette vapors on human reproductive health, human development, or the functioning of the organs of the male and female reproductive systems. Data on the effects of the exposure were derived mainly from animal studies, and they show that e-cigarettes can affect fertility. Here, we review recent studies on the effects of exposure to e-cigarettes on facets of morphology and function in the male and female reproductive organs. E-cigarettes, even those which are nicotine-free, contain many harmful substances, including endocrine disruptors, which disturb hormonal balance and morphology and the function of the reproductive organs. E-cigarettes cannot be considered a completely healthy alternative to smoking. As is true for smoking, deleterious effects on the human reproductive system from vaping are likely, from the limited evidence to date.

## 1. Introduction

Cigarette smoking is a major public health problem in many countries throughout world. The harmful effects of smoking on human health are well documented in both in vivo and in vitro studies; smoking is known to be a major risk factor for respiratory diseases, cardiovascular disorders, hormonal imbalance, cancer, and disturbances of the reproductive system [1,2,3,4,5,6,7]. A decrease in semen parameters, with a higher percentage of sperm with abnormal morphology, is observed in smoking men. Paternal smoking was also suggested to decrease in vitro fertilization (IVF) success rates [8]. Additionally, tobacco use by women during pregnancy can result in gestational hypertension, chorioamnionitis, preterm birth, impaired fetal lung function, reduced fetal growth, and a higher risk of perinatal and obstetric complications [9,10].

The relationship between cigarette smoking and male and female fertility was widely investigated [3,11,12]. Reduced total sperm count, sperm parameters, and motility were observed [13]. Additionally, an alteration in the ratios between protamines 1 and 2 (PRM1/PRM2) and aberrations in the histone-to-protamine ratios correlate with male infertility [14,15]. Protamine transcript ratios can also serve as a marker for male fertility [16].

The effect of smoking on female fertility is difficult to discern. However, it was estimated that cigarette smoking can affect conception, ovarian follicular dynamics, gamete mutations, early pregnancy, and assisted reproductive technology (ART) outcomes [13,17,18]. There are results indicating that infertility is increased and fecundity is decreased among smokers, compared to nonsmokers. Moreover, the time of conception increases in smokers, especially in women who continued to smoke close to the time, and the chances of becoming pregnant are cut almost in half each month [18,19]. It was also observed that the risk of early natural menopause is higher in smoking women [20], and that smoking can enhance ovarian follicular depletion [21]. This review presents recent studies of the effects on exposure to e-cigarettes on facets of the morphology and function of male and female reproductive organs.

## 2. E-Cigarettes

With the increasing awareness of the health dangers associated with cigarette smoking and the restrictions on public smoking, interest in electronic cigarette use is increasing. The battery-operated electronic nicotine delivery system was first developed by the Chinese pharmacist Hon Lik as an electronic alternative to traditional cigarettes in 2003 for Golden Dragon Holdings (now known as Ruyan), which started exporting in 2005 and 2006 [22]. Since then the popularity of electronic cigarettes increased, particularly among young adults. Unlike tobacco products, the use of e-cigarettes was not age-restricted for a number of years. Today, many states and countries prohibit the sale of e-cigarettes to minors. In some cases, local authorities forbid their use in public places [23].

E-cigarettes deliver a nicotine-containing aerosol (vapor) by heating a so-called e-fluid enclosed in a cartridge. The nicotine content of the e-fluid of e-cigarettes varies; some fluids are free from nicotine. The aerosol is inhaled by users and then exhaled into the environment. It was estimated that an individual puff contains 0–35 μg of nicotine [24,25]. Nevertheless, studies indicated that the levels of all toxicants per puff are much lower in e-aerosols than in smoke from conventional cigarettes [26,27,28,29]. The e-liquid in e-cigarette cartridges typically contains two main inhaling aerosolized humectants, such as propylene glycol (PG) and glycerol (vegetable glycerine); their proportions vary depending on the brand and the manufacturer [30]. Propylene glycol and vegetable glycerine are regarded as nontoxic when delivered orally. However, when e-liquids are heated, a number of harmful compounds—including formaldehyde, acetaldehyde, methylglyoxal, acrolein, acetone, benzaldehyde, and the so-called BTEX compounds (benzene, toluene, ethylbenzene, and xylenes)—are found in the inhaled vapor. These are produced primarily by the oxidation and thermal decomposition of the two main components of e-fluid, glycerol and PG [31]. Moreover, e-liquids contain a range of flavorings, such as fruit and sweet flavors sold under names such as “Candy Corn”, “Chocolate Fudge”, and “Berry Splash”; in 2014, and the number of e-liquid flavors exceeded 7500, and it continues to increase [32,33]. In addition to various high levels of PG, glycerol, and nicotine, e-liquids can contain nanoparticles, metals such as lead, chromium, tin, silver, nickel, copper, aluminum, cadmium, and mercury, tobacco-specific nitrosamines (TSNAs), hydroxycarbons, polycyclic aromatic hydroxycarbons (PAHs), phenols, aldehydes, and pesticides [34,35,36]. Analysis of 28 e-cigarette liquids identified 141 flavor chemicals—most frequently vanillin, ethyl maltol, ethyl vanillin, and menthol [37]. In another study, aldehydes such as benzaldehyde and vanillin, both of which can cause respiratory irritation, were identified in 30 e-cigarette fluids [38].

Flavor chemicals are found in e-fluids for every type of e-cigarette on the market. These represent a largely unrecognized potential hazard of electronic cigarettes, and they cannot be recognized as safe for inhalation [34]. The exhaled vapor can, thus, be regarded as a new source of pollution and toxins in the environment (American Nonsmokers’ Rights Foundation 2019).

While the effect of conventional cigarette smoking on reproduction is well documented (Figure 1), the scientific evidence for the effects on the male and female reproductive systems of exposure to e-cigarette aerosol is, to date, limited. Data of the effect of exposure to e-cigarettes are mainly derived from animal studies. Vaping—the use of e-cigarettes—is widely advertised as safer than cigarette smoking.

## 3. Male Reproductive System

The first publication dealing with the effect of e-cigarette refill liquids, with and without nicotine, on rat testes was published by El Golli et al. (2016) [39], in a study performed in an animal model (Figure 2). In the experiment, Wistar rats weighing 160 ± 20 g were exposed to electronic cigarette refill liquid for four weeks through daily intraperitoneal injections. The results showed that e-cigarette refill fluids, regardless of whether they contained nicotine or not, induced oxidative stress, leading to a significant increase in the activity of antioxidant enzymes such as superoxide dismutase, catalase, and glutathione *S*-transferase in rat testes. Histopathological changes in testis morphology were observed, including premature sloughing of germ cells from the seminiferous epithelium and disorganization of the tubular contents of the testes [39]. The morphology and the function of the testes are under the broad control of hormones, mainly androgens. Exposure to e-liquid, with or without nicotine, resulted in a marked decrease in circulating testosterone levels (by 50% and 30%, respectively) because of a decrease in the messenger RNA (mRNA) expression of two key steroidogenesis enzymes, cytochrome P450scc and 17β-Hydroxysteroid dehydrogenases (17β-HSD). Sperm collected from the epididymis cauda revealed a significant decrease in sperm count and viability [39].

A continuation of the study in the same animal model was performed to determine the effect of e-liquids with or without nicotine on rat epididymides [40]. Exposure to e-liquid induced a significant decrease in the number of epididymal spermatozoa in liquids with and without nicotine. Interestingly, the sperm numbers in rat treated with the fluid without nicotine were lower than in the other group with nicotine, at 32.3 ± 3.0 million/mL and 38.4 ± 0.9 million/mL, respectively. Similar results were obtained when comparing the viability of sperm for exposed rats, at 27.0% ± 4.6% and 42.8% ± 5.1% of live sperm. A morphological study indicated a significant increase in the percentage of sperm with abnormalities, especially in the rats exposed to the nicotine-free liquid vs. the liquid with nicotine vs. control, at 43.0% ± 1.0% vs. 30.2% ± 1.8% vs. 24.0% ± 0.9%. After four weeks of treatment, a notable decrease in circulating testosterone levels was noted in both experimental groups. The e-liquid also induced effects on oxidative status in the epididymides of the experimental rats.

A similar study was performed by Vivarelli et al. [41], using seven-week-old Sprague-Dawley rats to test a new-generation e-cigarette device, which was adjusted to one of the lowest possible voltage values and filled with a nicotine-free liquid with 50% *v*/*v* propylene glycol (PG) and 50% vegetable glycerine (VG). The seven rats in the experimental group were exposed to the vapor from the e-cigarettes for 28 days, 3 h daily, as 11 cycles of two puffs. The e-cigarette cartridge was filled with nicotine-free liquid composed of PG and VG flavored with 10% red fruit aroma [41]. The chemical analysis allowed toxic aldehydes such as formaldehyde, acetaldehyde, and acrolin to be detected in the vapor. In accordance with data from the literature [42,43], the exposure of male animals to aldehydes resulted in alterations in steroidogenesis enzyme activity. The inhibition of the expression of two key enzymes included in steroid synthesis, 3β-HSD and 17β-HSD, was observed after exposure to e-cigarette vapor. As a consequence, the activity of the testicular marker enzymes sorbitol dehydrogenase (SDH) and glucose-6-phosphate dehydrogenase (G6PDH) was significantly impaired, while that of the marker of tissue damage lactate dehydrogenase (LDH) slightly increased. Higher levels of reactive oxygen species (ROS) were noted in the testes of the exposed animals, together with a significant increase in protein carbonyl formation and lipid peroxidation products; alterations were also seen in antioxidant and detoxifying enzymatic systems [41]. Although the morphology of the testes was not studied, the results suggest that the structure of the testis and the cellular organization of the seminiferous epithelium underwent changes. In conclusion, the authors emphasized that exposure to vapor from a low-voltage, nicotine-free e-cigarette liquid induced the disruption of enzymes involved in steroidogenesis, as well as of those linked to the activity of the seminiferous epithelium, suggesting impairments of the reproductive system [41].

No results were published on the effects of e-cigarette use on spermatozoa. Nevertheless, Helen O’Neill suggested at the British Fertility Society Conference 2017 in Edinburgh that electronic cigarettes could damage men’s fertility through toxic chemicals in the flavorings [44]. She was presenting findings of an experiment in which men’s spermatozoa were exposed to cinnamon and bubble-gum flavors introduced into the medium. The concentration of the flavors utilized in the experiment was similar to the average intake for casual and more habitual e-cigarette users. Sperm samples were taken from 30 male candidates who were part of an in vitro fertilization (IVF) program. The results indicated that cinnamon vape flavors can significantly reduce the motility of spermatozoa, causing the cells to move more slowly. In the other part of the experiment performed by O’Neill (2017) [44] and her students, male mice were exposed to cinnamon flavor and another of the most popular flavors (bubble gum) to determine their effect on the seminiferous epithelium. It was found that the bubble-gum flavor damaged the germ cells in the testes of mice. The study concluded that the cinnamon flavor affects spermatozoa motility, whereas the bubble-gum flavor destroyed the cells in the testes responsible for producing sperm [44].

In a study by Wawryk-Gawda et al. [45], the effects on fertility of exposure to smoke or e-vapor were studied in male rats. Both smoke and e-vapor exposure produced morphological and functional changes in the seminiferous epithelium, such as vacuolization, reduction of spermatogenesis, and increased apoptosis of spermatogonia and spermatocytes. Additionally, slight changes in sperm morphology were observed. The changes were more pronounced in rats exposed to smoke from traditional cigarettes, and the male reproductive organs were slightly less affected by vapor [45].

Studies based on effects of e-cigarettes on the male reproductive system mostly dealt with animal models. However, the data above suggest that vaping leads to pathological alterations of the cells, tissues, and organs of the male reproductive system. Male teenagers should be aware that not only can nicotine-based e-cigarettes affect their reproductive system and future fatherhood, but that nicotine-free and flavored e-cigarettes can also have the same effect.

## 4. Female Reproductive System

The effect of cigarette smoking on female fertility is well documented, and there are data that indicate that active long-term smoking at high intensity significantly reduces fecundability and has harmful effects during pregnancy [46,47]. Currently, there is a major knowledge gap regarding the effects of inhaling and exhaling e-cigarette vapor on folliculogenesis and gamete competence, as well as regarding the e-cigarette exposure of pregnant women and the embryo/fetus in prenatal life, and the health risk for the mother [48]. Many women attempt to quit smoking when they are pregnant or plan to be. Pregnant women and women of child-bearing age often use electronic cigarettes, due to the opinion that vaping is less harmful than conventional cigarettes [25].

As mentioned above, the effect of exposure to e-cigarettes on women’s reproductive health is not yet determined, and the published data were mainly from studies on animal models. To examine the effects of e-cigarettes on pregnancy initiation, second-generation fetal reproductive health, implantation, and the health of future offspring, pre-clinical studies were performed using C57BL/6J mice [49]. These mice were exposed to e-cigarette aerosols using the SCIREQ inExpose computerized whole-body inhalation system for animal models. They were exposed 3 h/day with two puffs/min, with a puff duration of about 2 s. The fluid contained a propylene glycol and vegetable glycerin mixture with 24 mg/mL nicotine and no flavorings. The vapor was generated at 245 °C. Females were mated and, from that day, in parallel with control mice, they were exposed to e-cigarettes five days a week for four months, leading to a fertility trial. To study the effects of e-cigarettes on the implantation process, virgin mice were exposed to vapor for four weeks before the mating, and, after that, the pregnant mice were exposed every day to e-cigarettes and euthanized five days after the presence of the copulatory plug (day 5.5). The exposure of pregnant mice to vapor resulted in a mildly decreased offspring number per litter, and no differences in the weight of pups were observed. Dams exposed to e-cigarette vapor showed a delay in the onset of their first litter by three to four days compared to the control animals [49]. The exposure of the mice to e-cigarette vapor before mating and next during pregnancy caused a delay in embryo attachment. The morphology of the implantation sites in experimental mice exhibited disorganization with hemorrhagic blood cells, and only one embryo implant for ten dams was noted by day 5.5 of pregnancy.

The embryo’s development to the preimplantation blastocyst stage is concomitant with uterine differentiation, producing the receptive state—the “implantation window”—which is crucial for successful embryo attachment and implantation [50]. The implantation process is complex and involves the interplay of numerous signaling molecules [51]. The next step in the study of Wetendorf et al. (2019) [49] was to assess how exposure to e-cigarettes delayed implantation. Analysis of an RNA microarray was performed on sham and e-cigarette-exposed uteri at pseudopregnancy day 4.5. Transcriptome analysis indicated that exposure to e-cigarettes modulated the expression of genes necessary for uterine receptivity, such as integrin, prostanoid biosynthesis, proliferation, Janus kinase (JAK), and chemokine signaling. Additionally, the gene encoding the tight junction protein claudin 10 (CLDN10) in the surface epithelium and glandular epithelium in the stroma of the uterus was upregulated, with increased RNA levels [49].

The next step in the study assessed fetal health after exposure to vapor. Litters from exposed mice dams were aged 8–12 weeks and were then mated. The in utero exposure of the dams did not have an effect on reproductive efficiency or F2 generation health. On the other hand, in males, a slight reduction in fertility and F2 offspring weight and number was observed, yet there were no differences in testis morphology, sperm count, motility, or seminal vesicle weight in adult males exposed to e-cigarettes in utero, compared to sham male mice. In contrast, the exposure in utero of female mice resulted in decreased weight gain [49]. The results clearly indicated that e-cigarettes negatively affected the implantation of the embryo and led to an abnormal course of pregnancy, affecting the health of the progeny exposed in utero. If applied to women, these results would indicate that e-cigarette use in reproductive age can directly and negatively affect conception and have harmful effects on the embryo and fetus [49].

In next experiment, pregnant C57BL/6J mice were exposed to e-cigarette nicotine vapor [52]. Although the purpose of this study was to determine the effect of exposure to e-cigarettes during periods of intense brain growth, the effect on pup weight was also determined. Joyetech 510-T e-cigarettes with 510-T tank cartridges, atomizer, and battery were used. The fluid contained 2.4% nicotine in propylene glycol (PG) or 0% nicotine/PG. Termed pregnant mice from day 15 to 19 were exposed to vapors with or without nicotine content, once a day for approximately 20 minutes. Pups were also exposed in the same condition from postnatal day 2–16. The nicotine exposure was calculated as 2.1 mg/day. On the first day of postnatal life, the mean weight of pups exposed to 0% nicotine/PG vapors in prenatal life was significantly less than both the 2.4% nicotine/PG and the untreated mice. The mean weight of pups at day 7 of postnatal exposure to nicotine vapor was significantly lower than that of untreated control mice. The mean weight of mice exposed to vapor without nicotine was also less than that of untreated mice. To assess the serum nicotine content, the level of cotinine, a major proximate metabolite of nicotine and a biomarker for systemic nicotine absorption [53], was measured. It was shown that the mean level of cotinine in pup serum at the end of the 14 days of postnatal exposure was 23.7 ± 4.2 ng/mL in the 2.4% nicotine/PG mice, 2.8 ± 0.3 ng/mL in the 0% nicotine/PG mice, and 1.0 ± 0.001 ng/mL in the untreated mice. It was very interesting that the body weight of pups exposed to vapor without nicotine was lower than that of both the untreated pups and the pups exposed to nicotine vapor. Although the mice were exposed to vapors only once a day for about 20 min, the exposure had an unfavorable effect on their weight. In relation to the main purpose of the work, the results indicated that mice’s exposure to nicotine-containing vapor during a period of rapid brain growth can cause persistent behavioral changes [52].

The impact of e-cigarette exposure on weight gain and postnatal lung growth in neonatal mice was also assessed by the same group of researchers [54]. In this experiment, pups born from C57BL/6J mice were exposed to e-cigarette puffs, beginning at 24 h of life, receiving 1.8% nicotine/PG or 0% nicotine/PG once a day for days 1 and 2 of life, and then twice a day from days 3–9 of life. The experimental mice and the untreated control mice were weighed daily. At ten days of life, mice exposed to 1.8% nicotine vapor showed a 13.3% decrease in total body weight, while those exposed to the 0% nicotine vapor showed an 11.5% decrease in total body weight—significantly less than the untreated mice. Cotinine levels were also measured in the plasma and urine of the exposed neonatal mice at 10 days of life. The highest level of the marker was detected in mice exposed to 1.8% nicotine/PG, with a mean plasma level of 62.3 ± 3.3 ng/mL and a mean urine level of 892.5 ± 234 ng/mL, while, in mice exposed to 0% nicotine/PG, the levels were less than 5ng/mL and less than 10 ng/mL, respectively. A significant association between total body weight and plasma cotinine level was also found. It was additionally shown that postnatal exposure to e-cigarettes containing nicotine can lead to a decrease in alveolar cell proliferation, thereby producing a modest inhibition in postnatal lung growth [54].

The effects of chronic exposure to e-cigarette aerosols on early development was studied in a rat model, which was in fact the first study in an animal model [55]. In that experiment, timed-pregnant Sprague-Dawley rats and then their pups were exposed to the vapor of e-cigarettes without nicotine (pair-fed juice; the “juice” group) and with nicotine (juice + nicotine; the “nicotine” group). The control animals were exposed to pair-fed control (the “control” group). To determine the effect of vaping on the weight of newborns, pregnant dams underwent vaping treatment for 3 h a day, 5 days/week, from gestational day 5 until day 20, two days prior to parturition. The pregnant dams were treated with vapor for 2 h/day, 5 days/week from gestational day 5 until day 21, and after birth on day 22. The mother and pups again received vapor treatment from postnatal day 4 until day 9. Mass spectrometric analysis was performed to detect vapor compounds within the vaping chamber. Together with propylene glycol, glycerol, and nicotine, 17 other aerosolized chemicals (including formic acid, butyl ester, and acetic acid) were detected. Fetal and pup growth was measured on gestational day 20 and on postnatal day 10, after the last vaping episode. There were no differences in mean fetal weight between the control and juice groups, but the weight significantly decreased in the nicotine group. The fetal crown–rump length was also significantly decreased. The same trend was observed for weight and crown–rump length of pups exposed in prenatal and postnatal life. Both parameters were lower in pups of the nicotine group. The study indicated that chronic exposure to vapor containing nicotine during the early stages of prenatal life can exert harmful effects and can lead to a reduction in weight and crown–rump length in the exposed offspring. The growth restriction may be caused by a significant decrease in uterine artery and fetal umbilical cord artery blood flow in animals exposed to nicotine vapor [55].

All the studies presented here that involved animal models indicated that exposure of the fetus during intrauterine life or of pups postnatally to e-cigarette vapors containing nicotine caused harmful effects, including decreases in weight and length. There are not yet any studies assessing the reproductive effects of e-cigarette use by women during pregnancy [56]. However, the use of e-cigarettes can provide levels of nicotine and its metabolites that are similar, or even higher, than those provided by traditional cigarettes, with similar systemic retention [57]. The results of a number of literature reviews indicated that the use of electronic nicotine delivery systems (ENDS) by pregnant women is not safe for fetuses [56,58,59,60]. However, it was observed that electronic cigarette usage is increasing rapidly in pregnant women, as well as in women of reproductive age, as it is thought to be healthier than smoking and useful as an aid for reducing and stopping smoking. At this time, the effects of e-cigarettes on human development are completely unknown [61]. There is, thus, a suggestion to promote awareness among the public and healthcare providers of the risk and benefits of using e-cigarettes during pregnancy [62]. Clear, evidence-based practice guidelines on e-cigarette use during pregnancy also need to be prepared by world healthcare organizations. This is important because, as animal models indicate, exposure to e-vapor during critical developmental periods—especially in utero—can impair organ development and lead to organ damage [63].

## 5. Conclusions

Data on the effects of e-cigarettes on reproductive systems and organs are very scarce. Most research used an animal model, with these studies clearly indicating that the use of e-cigarettes can disturb the seminiferous epithelium and sperm morphology. It can further exert harmful effects during the process of implantation and, particularly during pregnancy, can lead to various pathologies in the offspring, as presented in Figure 1. It is hard to explain the effect of vaping on human reproductive organs, but the data on animal models are alarming. E-cigarettes are used with increasing frequency by teenagers, who are potential future parents. E-cigarettes, even when nicotine-free, include many harmful substances (including endocrine disruptors) that disturb the hormonal balance and negatively affect the morphology and function of the reproductive organs. E-cigarettes cannot be considered a completely healthy alternative to smoking. As is true for smoking, deleterious effects on the human reproductive system from vaping are likely, from the limited evidence to date. It is of note that the flavors found in e-fluids are extremely diverse, and they are frequently more toxic than nicotine alone, which was also demonstrated on animal tissue. The effects of e-fluids with or without nicotine on male reproductive system are presented in Figure 2. The increasing popularity of e-cigarette smoking, especially by teenagers, means that more researched involving human tissues and organs is needed, including an analysis of the health effects of switching from tobacco smoking to vaping. It is, of note, that vaping cannot be used to dissuade smokers from using vaping to stop smoking. The evidence is very minimal; however, such as it is, it supports vaping as safer than smoking, although not harm-free.

## Figures and Tables

**Figure 1 ijerph-17-06152-f001:**
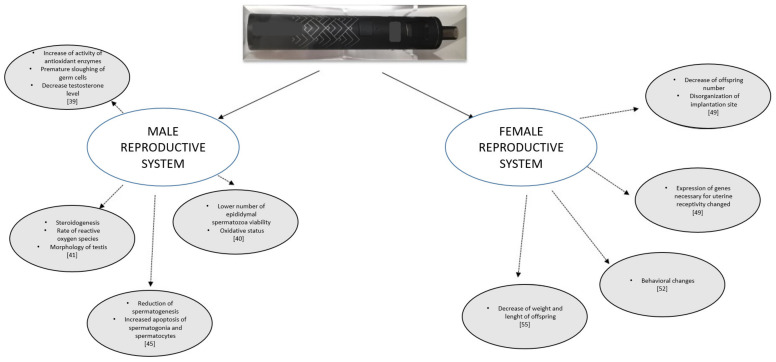
The influence of vapor components of e-cigarette on male and female reproductive system.

**Figure 2 ijerph-17-06152-f002:**
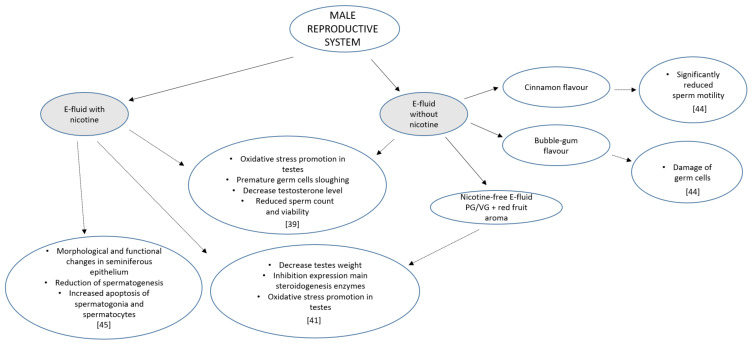
The effect of e-fluid with and without nicotine on male reproductive system.

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
