# Peer review of "The Effects of E-Cigarette Vapor Components on the Morphology and Function of the Male and Female Reproductive Systems: A Systematic Review"

_ijerph, 2020, doi:10.3390/ijerph17176152_

Round 1

Reviewer 1 Report

The aim of this review was the effect of the use of the E-cigarette on the male and female reproductive systems.

This review is well informative on these effects of the E-cigarette practice. The theme is original as few recent reviews are available on this particular subject.

Other comments:

In the figure 1: testosterone instead of testosteron.

The reference of this figure in the text should be in the conclusion and not in the introduction.

In paragraph 4, around the line 65 of this paragraph : nicotine instead of cotinine.

Ref45: the link is not correct

Ref 62 : E1560-E1569.

Author Response

Dear Reviewer #1:

Thank you for your advice and constructive comments concerning our manuscript entitled, “The influence of vapour components of E-cigarette on morphology and function male and female reproductive systems – A systematic review”. We have carefully considered your suggestions, and have revised our manuscript accordingly; we hope that these changes meet with your approval.

Reviewer #1:

The aim of this review was the effect of the use of the E-cigarette on the male and female reproductive systems.

This review is well informative on these effects of the E-cigarette practice. The theme is original as few recent reviews are available on this particular subject.

Other comments:

In the figure 1: testosterone instead of testosteron.

The reference of this figure in the text should be in the conclusion and not in the introduction.

In paragraph 4, around the line 65 of this paragraph : nicotine instead of cotinine.

Ref45: the link is not correct

Ref 62 : E1560-E1569.

Reviewer #1

  1. In the figure 1: testosterone instead of testosterone
  2. The reference of this figure in the text should be in the conclusion and not in the introduction.

(response)

We wish to thank Reviewer for his/her useful comments.

  1. The name of the hormone in Fig. 1 was spelled correctly - testosterone
  2. According to you suggestion, the Fig. 1 reference now is included in conclusions.

Reviewer #1

In paragraph 4, around the line 65 of this paragraph : nicotine instead of cotinine.

(response)

Thank you for your comment, which may improve the paper. In the sentence an information on cotinine was added: To assess the content of nicotine in serum, the cotinine level - major proximate metabolite of nicotine - as biomarker for systemic nicotine absorption [Benowitz N L 1996], was measured.

Benowitz N L: Cotinine as a biomarker of environmental tobacco smoke exposure. Epidemiol Rev 1996;18(2):188-204. doi: 10.1093/oxfordjournals.epirev.a017925

Reviewer #1

Ref45: the link is not correct

Wawryk-Gawda E, Zarobkiewicz MK, Chłapek K, Chylińska-Wrzos P, Jodłowska-Jędrych B: Histological changes in the reproductive system of male rats exposed to cigarette smoke or electronic cigarette vapor. Toxicological & Environmental Chemistry 2019, 101(7-8):404-419.(response) Ref. No. 45 has been corrected: Wawryk-Gawda E, Zarobkiewicz MK, Chłapek K, Chylińska-Wrzos P, Jodłowska-Jędrych B: Histological changes in the reproductive system of male rats exposed to cigarette smoke or electronic cigarette vapor. Toxicological & Environmental Chemistry 2019, 101(7-8):404-419.(response)

Reviewer #1

Ref 62 : E1560-E1569.

(response)

Pages in Ref. 62 are corrected: E1560-E1569

We would like to thank the Reviewers for helpful comments and criticisms. We believe that our revised manuscript is now more balanced and better represents our work. We hope that this revised manuscript is now acceptable for publication in International Journal of Environmental Research and  Public Health.

Reviewer 2 Report

I thank the authors for the chance to review their paper. Some significant work went into compiling the information, and it is an important topic. However I think it needs more work before it should be published. 

Minor problems include:

a. the English. I admire anyone who can write a paper in a language not their own, but it does need improving. The English phraseology is not quite right in many places. e.g. "adolescents OF both sexes", "attemptING to quit" both in the abstract. The whole paper has similar examples and needs going through carefully just to make it more readable. Also wrong words being chosen. e.g. "exposition" for "exposure" (abstract), "relief" for "refill" and "alternations" for "alterations", p5. Again, these are just examples.

b. References. Glaringly inadequate are refs 22 and 44 - but also when I followed the link for ref 45 it did not work. Perhaps they all need another check. References cited in the text all need a reference number by them.

c. Some carelessness with reporting e.g µ where µg was meant and talking about "both groups" (El Golli paper, p5) when it would have been much better to say "using e-liquids both with and without nicotine". Things which are not immediately clear interrupt the flow for the reader and detract from the message. It needs a careful read though to make sure that what you are saying will be crystal clear to a first time reader. 

Importantly, and aside from all these quibbles, I remained unimpressed by the general arrangement of the review. It was a straightforward linear summary of other people's results. It did not make any attempt at analysis or classification other than between male and female reproductive systems.

I would have liked to see it rearranged so that it was easy to see which effects were linked to nicotine, which to the carrier liquids and which to individual flavours. Arranging it that way would also make it easy to look at the doses used in the experiments and get some feeling for whether the effects seen were likely to translate across to people, bearing in mind the different doses, relative to human exposures and exposure routes. 

Of interest to anyone in the e-cigarette field is the question of the effects you might get from vaping, but also which health effects of tobacco smoking will be reduced by a switch to vaping. And of course there could be new effects. Thus it is important that, where possible, a review such as this gives one a sense of how the reviewed papers answer these questions with regard to the reproductive effects of vaping/nicotine. I would also like to see more understanding of the limitations of these model systems. At the moment the review does not leave me with anything more than "these different problems have been suggested in the literature", which is of limited utility. 

It is quite close to being a very useful review, so it would be worth making the effort to improve it.  

Author Response

Dear Reviewer #2

Thank you for your advice and constructive comments concerning our manuscript entitled, “The influence of vepour components of E-cigarette on morphology and function male and female reproductive systems – A systematic review”. We have carefully considered your suggestions, and have revised our manuscript accordingly; we hope that these changes meet with your approval.

Reviewer #2:

  1. the English. I admire anyone who can write a paper in a language not their own, but it does need improving. The English phraseology is not quite right in many places. e.g. "adolescents OF both sexes", "attemptING to quit" both in the abstract. The whole paper has similar examples and needs going through carefully just to make it more readable. Also wrong words being chosen. e.g. "exposition" for "exposure" (abstract), "relief" for "refill" and "alternations" for "alterations", p5. Again, these are just examples.

(response)

We wish to thank the Reviewer for his/her useful comments.

We have improved English based on the professional proofreading.

Reviewer #2:

  1. References. Glaringly inadequate are refs 22 and 44 - but also when I followed the link for ref 45 it did not work. Perhaps they all need another check. References cited in the text all need a reference number by them.

(response)

To organize The References EndNote reference management software package was used. Ref 22 and Ref 44 were corrected manually

Ref 22:

van Dulken S. The patent for e-cigarettes. http://stephenvandulken.blogspot.com/2014/01/the-patents-for-e-cigarettes.html

Ref. 44

O’Neill H.: Effect of Electronic-cigarette flavourings on (I) human sperm motility, chromatin integrity in vitro and (II) mice testicular function in vivo. http://srf-reproduction.org/wp-content/uploads/2017/01/Fertility-2017-Final-Programme-and-Abstracts.pdf

Reviewer #2:

  1. Some carelessness with reporting e.g µ where µg was meant and talking about "both groups" (El Golli paper, p5) when it would have been much better to say "using e-liquids both with and without nicotine". Things which are not immediately clear interrupt the flow for the reader and detract from the message. It needs a careful read though to make sure that what you are saying will be crystal clear to first time reader. 

(response)

We wish to thank the Reviewer for his/her useful comments. We have changed the units where it was necessary. We have highlighted the changes in the text in red colour.Additionally, we have improved the text, according to your suggestions (p5- instead “both groups” we used using e-liquids both with and without nicotine".

Importantly, and aside from all these quibbles, I remained unimpressed by the general arrangement of the review. It was a straightforward linear summary of other people's results. It did not make any attempt at analysis or classification other than between male and female reproductive systems.

(response)

Thank you for your constructive comment. We have focused on the influence of e-cigarettes on male and female reproductive system due to the fact, that vaping is more and more popular among young people.Teenagers are extremely important group as potential future parents. As we documented in our review, vaping affects both male and female reproductive system, including morphology of spermatozoa. Furthermore, e-cigarette using leads to pathological alterations of embryos. The subject seems to be extremely unique and we decided to treat it as a separate review. Other systemic effects of e-cigarettes using have been described by us in another review which is in process. We wanted to emphasize in the current the importance of being aware by teenagers and young adults of harmful effects of vaping, which, unfortunately, becomes more and more popular.

I would have liked to see it rearranged so that it was easy to see which effects were linked to nicotine, which to the carrier liquids and which to individual flavours. Arranging it that way would also make it easy to look at the doses used in the experiments and get some feeling for whether the effects seen were likely to translate across to people, bearing in mind the different doses, relative to human exposures and exposure routes.

Thank you for the suggestion. We have formed additional figure, where we have shown the effects of e-cigarettes with and without nicotine and separate flavours.

Of interest to anyone in the e-cigarette field is the question of the effects you might get from vaping, but also which health effects of tobacco smoking will be reduced by a switch to vaping. And of course there could be new effects. Thus it is important that, where possible, a review such as this gives one a sense of how the reviewed papers answer these questions with regard to the reproductive effects of vaping/nicotine. I would also like to see more understanding of the limitations of these model systems. At the moment the review does not leave me with anything more than "these different problems have been suggested in the literature", which is of limited utility.

Dear Reviewer, thank you for that comment. However, the field of the researches based on e-cigarettes effects on reproductive systems/organs are very scarce. More researches are based on animal models. It is hard to analyze and explain the effect of vaping on human reproductive organs. However, data on animal models are alarming. Additionally, the are no data regarding health effects of tobacco smoking by  switching to vaping on reproductive system. This is our limitation that we have added to manuscript.

We would like to thank the Reviewers for helpful comments and criticisms. We believe that our revised manuscript is now more balanced and better represents our work. We hope that this revised manuscript is now acceptable for publication in International Journal of Environmental Research and  Public Health.

Round 2

Reviewer 2 Report

Overall this paper is much improved. I find it a real pleasure to read now, so the English editing was very good. 

I also appreciate the addition of Figure 2, which summarises  the overall split into "effects of nicotine" and "effects of flavours/carrier liquids" nicely. 

I have a couple of minor corrections to suggest. 

A typo on p 11 (1st paragraph) in the reporting of Smith et al results (ref 52). The results at 2.4 ng/mL  were for mice exposed to 0% nicotine, not 5% nicotine. It did not make sense that the result of exposure to 5% nicotine would be ten times lower than that from exposure to 2.4% nicotine.

Also an editing glitch led to a word (explain) being retained  when it was supposed to be replaced by "determine". (p12 1st para  of Conclusion.I expect there may be more minor editing glitches such as unnecessary spaces etc that become evident once the markup is removed, but these can be dealt with by normal editing. 

 I do have one important change to suggest. In both the abstract and the conclusion the authors state that vaping cannot be considered a healthier alternative to smoking. This conclusion is not supported by the evidence in the paper.

There is only one paper reported on that directly compares effects of smoke exposure and effects of e-vapour exposure, and that one states that the effects of e-vapour were less than the effects of tobacco smoke. (p 9, ref 45).  I understand the author's wish to make a strong statement, but they need to make that strong statement true. I suggest something like  "E-cigarettes cannot be considered a completely healthy alternative to smoking.  As is true for smoking, deleterious effects on the human reproductive system from vaping are likely, from the limited evidence to date." 

In general, we do not yet know the extent to which these alarming animal results are reflected in real world effects on vapers. This paper justifiably raises the alarm, but should be worded such that it cannot be used to dissuade smokers from using vaping to stop smoking. The evidence is very minimal, but, such as it is, supports vaping as safer than smoking, though not harm free.  

Author Response

Dear Reviewer #2

Thank you for your advice and constructive comments concerning our manuscript entitled, “The effects of vapor components of E-cigarette on morphology and function male and female reproductive systems – A systematic review”. We have carefully considered your suggestions, and have revised our manuscript accordingly; we hope that these changes meet with your approval.

Reviewer #2

Overall this paper is much improved. I find it a real pleasure to read now, so the English editing was very good. 

I also appreciate the addition of Figure 2, which summarises  the overall split into "effects of nicotine" and "effects of flavours/carrier liquids" nicely. 

I have a couple of minor corrections to suggest. 

A typo on p 11 (1st paragraph) in the reporting of Smith et al results (ref 52). The results at 2.4 ng/mL  were for mice exposed to 0% nicotine, not 5% nicotine. It did not make sense that the result of exposure to 5% nicotine would be ten times lower than that from exposure to 2.4% nicotine.

(response)

We wish to thank the Reviewer for his/her useful comments. We have changed the numbers in the text according to Smith et al.

Reviewer #2

Also an editing glitch led to a word (explain) being retained  when it was supposed to be replaced by "determine". (p12 1st para  of Conclusion.I expect there may be more minor editing glitches such as unnecessary spaces etc that become evident once the markup is removed, but these can be dealt with by normal editing. 

(response)

Thank you for that constructive comment. We have improved the text. We have deleted „determine” from the text. We have also checked the spaces in the text and we have edited them.

Reviewer #2

 I do have one important change to suggest. In both the abstract and the conclusion the authors state that vaping cannot be considered a healthier alternative to smoking. This conclusion is not supported by the evidence in the paper.

There is only one paper reported on that directly compares effects of smoke exposure and effects of e-vapour exposure, and that one states that the effects of e-vapour were less than the effects of tobacco smoke. (p 9, ref 45).  I understand the author's wish to make a strong statement, but they need to make that strong statement true. I suggest something like  "E-cigarettes cannot be considered a completely healthy alternative to smoking.  As is true for smoking, deleterious effects on the human reproductive system from vaping are likely, from the limited evidence to date." 

In general, we do not yet know the extent to which these alarming animal results are reflected in real world effects on vapers. This paper justifiably raises the alarm, but should be worded such that it cannot be used to dissuade smokers from using vaping to stop smoking. The evidence is very minimal, but, such as it is, supports vaping as safer than smoking, though not harm free.  

(response)

Thank you for that comment. We have added and changed the last sentence of the abstract and conclusions according to Your suggestions.

We would like to thank the Reviewer for helpful comments and criticisms. We believe that our revised manuscript is now more balanced and better represents our work. We hope that this revised manuscript is now acceptable for publication in International Journal of Environmental Research and Public Health.